# AdaBoost Ensemble Approach with Weak Classifiers for Gear Fault Diagnosis and Prognosis in DC Motors

Syed Safdar Hussain *[ID] and Syed Sajjad Haider Zaidi [ID]

Department of Electronics & Power Engineering, PN Engineering College, Karachi, National University of Sciences and Technology (NUST), Islamabad 44000, Pakistan; sajjadzaidi@pnec.nust.edu.pk
* Correspondence: shussain.phdee16pnec@student.nust.edu.pk

**Abstract:** This study introduces a novel predictive methodology for diagnosing and predicting gear problems in DC motors. Leveraging AdaBoost with weak classifiers and regressors, the diagnostic aspect categorizes the machine's current operational state by analyzing time–frequency features extracted from motor current signals. AdaBoost classifiers are employed as weak learners to effectively identify fault severity conditions. Meanwhile, the prognostic aspect utilizes AdaBoost regressors, also acting as weak learners trained on the same features, to predict the machine's future state and estimate its remaining useful life. A key contribution of this approach is its ability to address the challenge of limited historical data for electrical equipment by optimizing AdaBoost parameters with minimal data. Experimental validation is conducted using a dedicated setup to collect comprehensive data. Through illustrative examples using experimental data, the efficacy of this method in identifying malfunctions and precisely forecasting the remaining lifespan of DC motors is demonstrated.

**Keywords:** DC motor; diagnosing; AdaBoost; weak classifiers; fault severity conditions; remaining useful life

## 1. Introduction

The reliable and safe operation of electrical machinery has been receiving more attention in recent years. A great deal of research has been conducted on diagnostics, the early identification of electrical machine defects, and the estimation of remaining life [1]. Within the realm of automotive engineering, the utilization of DC motors stands as a cornerstone, celebrated for their versatility, precise control capabilities, and robust torque output. Based on the basic ideas of electromagnetism, these motors use Lorentz force interactions to turn electrical energy into mechanical motion. They play important roles in many automotive applications [2]. Notably, in the context of vehicle propulsion, DC motors often assume the pivotal role of self-starters, facilitating the ignition process in internal combustion engines with seamless efficiency. The evolutionary trajectory of DC motors has been marked by significant milestones, prominently characterized by the transition from conventional brushed motors to advanced brushless DC motor variants [2]. The relentless pursuit of enhanced performance metrics, including elevated efficiency, extended operational lifespan, and reduced maintenance requirements, has fueled this paradigm shift. Brushless DC motors, in particular, have emerged as the vanguard of this technological revolution, leveraging sophisticated electronic control mechanisms to deliver unparalleled precision and reliability in automotive systems [3]. Amidst these advancements, the integrity and functionality of DC motor gear systems have emerged as focal points of concern within automotive engineering circles [4]. These gear systems, indispensable for torque transmission and speed regulation, are susceptible to a myriad of faults that can compromise their performance and longevity. Some common problems are gear wear, which is when material slowly breaks down because of frictional forces; misalignment from bad assembly or mechanical stress; and tooth damage from too much load or manufacturing flaws [4]. The ramifications of gear faults can be particularly pronounced in the context of starter DC motors, which

initiate the combustion process in internal combustion engines. The onset of gear wear, for instance, can manifest as diminished torque output and increased mechanical noise during operation, adversely affecting the engine's ignition process and overall performance [4]. Similarly, misaligned or damaged gear teeth can exacerbate these issues, leading to suboptimal engine starting capabilities and potential safety hazards. Given the critical role of DC motor gear systems in ensuring the reliable operation of automotive systems, the development of comprehensive diagnostic and maintenance strategies is imperative [4]. By leveraging advanced predictive maintenance techniques, such as condition monitoring and fault prognostics, automotive engineers can proactively identify and address potential gear faults before they escalate into critical failures. Additionally, ongoing research efforts aimed at optimizing gear design, material selection, and lubrication practices hold promise for further enhancing the resilience and durability of DC motor gear systems in automotive applications. Moving forward, diagnostics of electrical machinery encompass three main techniques: model-based, signal-based, and data-based [5]. It is possible to use signal-based techniques in the time, frequency, and time–frequency domains [6]. The classification of faults also makes use of artificial intelligence and pattern recognition techniques [7]. A variety of time–frequency transformations were applied, such as the Choi–Williams, Wigner–Ville, short-time Fourier, and undecimated wavelet transforms [8]. To identify and categorize nonstationary defects in permanent magnet AC machines at an early stage, these transformations were coupled with classification techniques, including the multiple discriminant-based classifier, the closest neighborhood classifier, and the linear discriminant classifier [9]. In contrast to prognosis, diagnosis has been the main focus of study. More research has been conducted in the recent literature on fault prognosis, which includes estimating remaining useful life (RUL) and projecting future states [10]. One frequently used prognostic method revolves around particle filters, especially when it comes to forecasting the RUL of rotary systems experiencing problems like cracks [11]. Moreover, similarity-based prognostic techniques play a role in estimating RUL. These methods involve examining the life patterns of machines derived from run-to-failure data found in training datasets [12]. Predicting the RUL of a test sample involves matching patterns against these established ones. Another notable approach to prognosis employs neural networks, particularly in forecasting the RUL of batteries [13]. Both similarity-based prognostication and neural networks are categorized as data-based methods. Another important category of prognostic methods is the model-based approach. Here, models are created using a mathematical approach, the physical installation of the equipment, or a combination of both [14]. This approach views faults as continuous variables, following either a definite or random rule in their development. Models such as the Time-Dependent Dielectric Breakdown (TDDB) model and the Hot Carrier model play crucial roles in estimating the lifespan of semiconductor devices [15]. Batteries also use these models to determine their state of charge. Another significant approach to fault prognosis is through the use of Hidden Markov Models (HMMs). These statistical models are highly effective due to their inherent dual statistical nature. HMMs have widespread applications across various fields, including speech recognition, hand gesture recognition, and text segmentation [16]. Moreover, they are instrumental in tasks like detecting and predicting tool wear, as well as monitoring bearing faults. HMMs also play a crucial role in classifying structural damage and tracking the wear of machine tools. Transitioning from the discussion on fault prognosis, recent advancements in fault diagnosis methodologies have introduced innovative approaches to address challenges in prognostics. A multihop graph pooling adversarial network based on distributed federated learning (MHGPAN-DFL) is one such approach that aims to accurately predict the remaining useful life (RUL) in complex systems [17]. Multihop graph pooling and adversarial transfer learning are two strategies that MHGPAN-DFL uses to deal with data dispersion and domain drift issues in a good way. The weighted modified conditional variational autoencoder (WM-CVAE) is a new tool that was created to help with class imbalance problems in data-driven methods. This method improves the quality of the data and the ability of fault diagnosis models

to learn, as demonstrated by tests on real-world industrial datasets [18]. The method also created health indicators and predictions for the remaining useful life (RUL) of systems with few failures, changing operating conditions, and little labeled data. This was made possible by using an LSTM autoencoder [19]. DC machines, valued for their simple operation and control, are widely used across various operations, including within the auxiliary systems of automobiles [20]. Despite not being directly involved in the traction system, their malfunction can lead to the overall failure of the vehicle. Over the past century, researchers have extensively studied DC motors, but only recently have they focused on diagnosing and predicting faults in these machines. Such research aims to ensure the continuous operation of these machines and estimate their remaining useful life (RUL). This study proposes an innovative prognosis method that uses weak classifiers in AdaBoost to predict future fault severity conditions in automotive starter DC machines with gear faults. Unlike traditional approaches, this method utilizes a combination of weak classifiers to enhance predictive accuracy and robustness. AdaBoost enables the effective integration and iterative performance improvement of multiple weak classifiers, leading to more accurate fault severity condition predictions. The primary contributions of this research lie in the novel methodologies developed for computing the parameters of the weak classifiers and devising the prediction algorithm. This approach provides a more reliable and effective tool for prognosticating faults in DC machines, thereby enhancing the reliability and longevity of automotive systems. The problem of training weak classifiers with Adaboost becomes more difficult when limited data is available. This paper explores the complexities of computing weak classifier parameters with sparse training data. An algorithm is suggested to determine the next most likely fault severity condition, addressing an important part of fault prognosis when limited data is available. This work provides a thorough discussion and a complete description of the implementation methods, going beyond simple preliminary findings. The goal is to give researchers and practitioners the tools and insights they need to deal with the challenges of training weak classifiers with Adaboost under data restrictions by offering a thorough overview.

## 2. Background

In modern industrial environments, ensuring the continuous and reliable operation of machinery is crucial for maintaining productivity and minimizing downtime [21]. Fault detection and diagnosis play a pivotal role in this context, allowing for proactive maintenance and timely interventions to prevent costly failures. To address these challenges, researchers and practitioners have turned to machine learning algorithms, which offer powerful tools for analyzing complex data and identifying patterns indicative of machinery faults [21]. This section explores various machine learning techniques employed in fault detection and diagnosis for industrial systems. The discussion encompasses popular algorithms such as K-Means clustering, linear regression, Long Short-Term Memory (LSTM) networks, Levenberg–Marquardt Neural Networks (LMNN), Random Forest Classifiers, Gradient Boosting Machines (GBMs), Extreme Gradient Boosting (XGBoost), Logistic Regression Classifiers, Support Vector Machines (SVMs) with linear kernels, Decision Tree Classifiers, and AdaBoost with weak classifiers. Each algorithm is examined in terms of its underlying principles, applications, and effectiveness in fault detection and diagnosis tasks. This exploration aims to provide insights into the diverse range of machine learning approaches available for fault detection and diagnosis in industrial settings. By understanding the strengths and limitations of each algorithm, practitioners can make informed decisions when selecting the most appropriate techniques for their specific application scenarios. This discussion also shows how important it is to use advanced machine learning techniques to make fault detection and diagnosis more reliable and effective. This will lead to better operational performance and cost savings in industrial systems [21].

### 2.1. K-Means Clustering

K-Means clustering, a popular unsupervised machine learning algorithm, divides data into K clusters. It aims to minimize the within-cluster variance by iteratively updating cluster centroids, as explained in Equation (1).

$$\underset{C}{\text{argmin}} \sum_{i=1}^{K} \sum_{x \in C_i} ||x - \mu_i||^2 \tag{1}$$

where $C$ is the set of clusters, $C_i$ is the $i$th cluster, $\mu_i$ is the centroid of cluster $C_i$, and $|| \cdot ||$ denotes the Euclidean distance. K-Means clustering is explored as a precursor to Operational Modal Analysis (OMA) in monitoring rotating machines used in energy generation [22]. By distinguishing suitable excitation conditions through statistical features, this approach enhances OMA accuracy, ensuring effective Structural Health Monitoring (SHM) and potentially preventing shutdowns or catastrophic failures, thereby safeguarding economic and social interests [22].

### 2.2. Linear Regression

Linear regression is a linear approach to modeling the relationship between a dependent variable and one or more independent variables [23]. The equation for a simple linear regression explained in Equation (2) with one independent variable $x$ is

$$Y = \beta_0 + \beta_1 X + \epsilon \tag{2}$$

The independent variable, $X$, influences the dependent variable, $Y$. The intercept is represented by $\beta_0$, the slope coefficient by $\beta_1$, and the error term by $\epsilon$. By employing linear regression analysis, vibration signals are examined to accurately forecast bearing failures. The signals are converted into the frequency domain, allowing for the computation of the integral of the curve, thus offering information on the remaining usable life (RUL) of the bearing [23]. It is important to note that the slope of the regression line is a very important clue; a positive slope means things are getting worse or will soon break down [23]. The proactive method described in [23] helps the industry reduce costs associated with unexpected periods of inactivity, urgent repairs, and replacing components. This technique improves operational efficiency and reliability.

### 2.3. Long Short-Term Memory

Long Short-Term Memory (LSTM) is a type of recurrent neural network (RNN) architecture designed to model sequential data [24]. It addresses the vanishing gradient problem of traditional RNNs by introducing gating mechanisms to control the flow of information. The LSTM unit consists of several gates, including an input gate ($i$), forget gate ($f$), output gate ($o$), and cell state ($C$). Activation functions control these gates and learn to regulate the flow of information through the cell. The Equations (3)–(8) governing LSTM operations are as follows:

$$f_t = \sigma(W_f \cdot [h_{t-1}, x_t] + b_f) \tag{3}$$

$$i_t = \sigma(W_i \cdot [h_{t-1}, x_t] + b_i) \tag{4}$$

$$\tilde{C}_t = \tanh(W_C \cdot [h_{t-1}, x_t] + b_C) \tag{5}$$

$$C_t = f_t * C_{t-1} + i_t * \tilde{C}_t \tag{6}$$

$$o_t = \sigma(W_o \cdot [h_{t-1}, x_t] + b_o) \tag{7}$$

$$h_t = o_t * \tanh(C_t) \tag{8}$$

The hidden state at time $t$ is denoted as $h_t$, the input at time $t$ is denoted as $x_t$, the sigmoid activation function is denoted as $\sigma$, element-wise multiplication is denoted as $*$, and the weight matrices and bias vectors are denoted as $W$ and $b$, correspondingly. These

parameters are learned during training. This method utilizes Long Short-Term Memory (LSTM) networks to advance the field of fault prognosis for rolling element bearings, which are crucial components in industrial setups [24]. The model achieves improved performance by directly importing raw time series sensor data, minimizing the need for feature engineering. LSTM, or Long Short-Term Memory, allows the model to surpass other models that rely on conventional characteristics, demonstrating its ability to accurately forecast faults [24]. Moreover, using unprocessed vibration data improves the model's capacity to make accurate predictions across different types of energy sources, including hydro and wind power turbines, demonstrating the robustness of LSTM in proactive, predictive maintenance applications [24].

### 2.4. Levenberg–Marquardt Neural Network

The Levenberg–Marquardt algorithm is an optimization method commonly used for training neural networks, especially in regression tasks [25]. Its purpose is to minimize the error function $E(w)$, which measures the discrepancy between predicted outputs $y_i(w)$ and actual outputs $t_i$ for a given set of input data $x_i$. The error function is defined in Equation (9):

$$E(w) = \frac{1}{2} \sum_{i=1}^{N} (y_i(w) - t_i)^2 \tag{9}$$

where $N$ represents the total number of training samples. The algorithm iteratively adjusts the parameters $w$ of the neural network model to minimize this error. It blends elements of gradient descent and Gauss–Newton methods to achieve optimization. At each iteration, the algorithm computes the Jacobian matrix $J(w)$, which contains the partial derivatives of the outputs concerning the parameters. This matrix offers insights into the local curvature of the error surface, aiding in determining the direction and magnitude of parameter updates. The update rule for adjusting the parameters is given by Equation (10).

$$w^{(k+1)} = w^{(k)} - \alpha (J(w^{(k)})^T J(w^{(k)})^{-1} \nabla E(w^{(k)})) \tag{10}$$

where $w^{(k)}$ denotes the parameter vector at iteration $k$, $\alpha$ represents the learning rate, and $\nabla E(w^{(k)})$ is the gradient of the error function. The algorithm inherently includes a regularization effect through its parameter update mechanism, which effectively constrains the model's complexity and reduces the risk of overfitting. By iteratively updating the parameters based on the gradient and curvature information provided by the Jacobian matrix, the Levenberg–Marquardt algorithm converges to the optimal parameter values, enabling effective training of neural networks for regression tasks.

### 2.5. Random Forest Classifier

Random Forest is an ensemble learning technique that creates several Decision Trees during training and produces the most frequent class (classification) or the average prediction (regression) of the various trees [26]. A randomly selected portion of the training data trains each tree inside the forest, and a randomly selected subset of features informs each split. A Random Forest Classifier's prediction is determined by aggregating each tree's majority vote. Denote the $i$th tree forecast as $T_i(x)$. The ultimate prediction of the Random Forest ensemble is determined by Equation (11).

$$\hat{Y}(x) = \text{mode}\{T_1(x), T_2(x), \ldots, T_n(x)\} \tag{11}$$

where $x$ represents the input data point. Random Forest stands out as a versatile and potent tool, significantly enhancing the reliability and efficiency of predictive maintenance strategies across diverse industrial applications [26]. Its robustness in handling complex datasets and ability to capture intricate patterns make it a valuable asset in ensuring the optimal performance and longevity of critical machinery like rolling bearings [26].

## 2.6. Gradient Boosting Classifier

Gradient Boosting is an ensemble learning strategy that constructs a powerful learner by iteratively incorporating weak learners, usually Decision Trees, into the ensemble [27]. In contrast to Random Forest, the Gradient Boosting model trains each tree on the residual errors of the preceding trees. Subsequent trees in the model prioritize the remaining faults. A Gradient Boosting Classifier determines its prediction by summing up the predictions of all the weak learners. Let $f_i(x)$ denote the prediction of the $i$th weak learner, then the final prediction of the Gradient Boosting ensemble is given by Equation (12).

$$\hat{Y}(x) = \sum_{i=1}^{n} f_i(x) \tag{12}$$

$n$ is the total number of weak learners, and $x$ represents the input data point. Gradient Boosting, a potent ensemble learning technique, is a versatile tool for various practical applications. This iterative model sequentially enhances prediction accuracy by iteratively creating new models [28]. Proposing a data-driven diagnostics and prognostics framework, researchers devised a precise data labeling approach for supervised learning, validating the framework using real vending machine data with SVM, RF, and Gradient Boosting Classifiers [29]. Results show the Gradient Boosting model achieved over 80% accuracy in diagnostics and outperformed traditional prediction models in two-stage prognostics, showcasing its efficacy in diagnosing and monitoring complex systems [29].

## 2.7. Extreme Gradient Boosting

Extreme Gradient Boosting (XGBoost) is a highly sophisticated implementation of the Gradient Boosting technique designed to prioritize speed and performance. The method employed is Gradient Boosting, which constructs a collection of weak learners, usually Decision Trees, and enhances the standard Gradient Boosting approach by integrating regularization and parallel processing [30]. Aggregating the predictions from several weak learners determines the prediction of an XGBoost classifier. The prediction of the $i$th weak learner is denoted as $f_i(x)$. The final prediction of the XGBoost ensemble is determined by Equation (13).

$$\hat{Y}(x) = \sum_{i=1}^{n} f_i(x) + \gamma \sum_{j=1}^{n} \Omega(f_j) \tag{13}$$

Let $n$ denote the total number of weak learners, $f_i(x)$ represent the prediction of the $i$th weak learner given input $x$, $\gamma$ be the regularization parameter, and $\Omega(f_j)$ be the regularization term. XGBoost was employed as one of the models for diagnostic analysis, demonstrating its efficacy in promptly and accurately detecting motor problems [31]. The results obtained from stratified K-fold cross-validation demonstrate that XGBoost is suitable for practical implementation in fault diagnosis for induction motors. In addition, a user-friendly graphical interface was created to simplify the execution of the diagnostic technique [31].

## 2.8. Logistic Regression Classifier

Logistic Regression is a popular classification algorithm used to model the probability of a binary outcome based on one or more predictor variables [32]. It estimates the probability that a given input belongs to a particular class by fitting a logistic function to the data. The Logistic Regression model predicts the probability of the positive class ($Y = 1$) using the logistic function, also known as the sigmoid function. The probability $p$ of class membership is given by Equation (14).

$$p(X) = \frac{1}{1 + e^{-\beta^T X}} \tag{14}$$

where $X$ represents the input features, $\beta$ is the vector of coefficients (including the intercept term), and $e$ is the base of the natural logarithm. This study looks at Logistic Regression (LR)

as a way to find problems in battery systems and shows that it is more accurate and easier to set up than other machine-learning-based methods [32]. LR demonstrates superior overall performance compared with k-NN, GNB, KSVM, and classical NN methods. However, this study did not consider various uncertainties present in real-world applications during model training and testing [32].

### 2.9. Support Vector Machine with Linear Kernel

Support Vector Machine (SVM) is a supervised learning algorithm for classification tasks. When using a linear kernel, SVM constructs a hyperplane that best separates the classes in the feature space. It maximizes the margin between the support vectors and the decision boundary [33]. In the case of SVM with a linear kernel, the decision function for classifying a new data point $X$ is given by Equation (15):

$$f(X) = \text{sign}(\sum_{i=1}^{n} \alpha_i y_i K(X, X_i) + b) \tag{15}$$

where $n$ is the number of support vectors, $\alpha_i$ are the Lagrange multipliers, $y_i$ are the class labels, $X_i$ are the support vectors, $K$ is the linear kernel function, and $b$ is the bias term. Detecting bolt loosening in rotating joints is a critical challenge in industrial settings [33]. This study employs Support Vector Machines (SVMs) for vibration-based detection in a custom sewer-cleaning vehicle transmission. The SVM proved highly effective, achieving a remarkable 92.4% accuracy [33]. Using information from accelerometers placed in strategic places, the SVM shows how reliable it is at finding faults in important industrial parts like rotating joints [33].

### 2.10. Decision Tree Classifier

A Decision Tree is a nonparametric supervised learning method for classification and regression tasks [34]. It recursively partitions the feature space into subsets based on the values of input features, with each partition represented by a tree node. Decision Trees are simple to understand and interpret, making them popular for exploratory data analysis [34]. A series of conditional statements based on input features represent the decision function of a Decision Tree Classifier. Let $X$ represent the input features and $T$ be the Decision Tree model. The prediction of the Decision Tree Classifier is given by Equation (16):

$$\hat{Y}(X) = T(X) \tag{16}$$

$T(X)$ represents the output of traversing the Decision Tree with input features $X$. An approach leveraging the Decision Tree algorithm is proposed for fault detection and diagnosis in grid-connected photovoltaic systems [35]. The Decision Tree algorithm accurately sorts faults into groups like free fault, string fault, short circuit fault, or line–line fault. This shows how well it works for finding and fixing faults in grid-connected photovoltaic systems [35].

### 2.11. Ensemble Learning Methods

Ensemble Learning Methods combine multiple base learners to enhance predictive performance. Techniques like Random Forests, Gradient Boosting Machines (GBMs), and Ensemble of Neural Networks leverage the diversity of individual models to reduce overfitting and improve generalization. The output of an ensemble classifier is obtained by aggregating the predictions of base learners, weighted by their respective performances.

### 2.12. AdaBoost with Weak Classifiers for Fault Severity Prediction

The proposed method utilizes a boosting algorithm known as AdaBoost, which combines multiple weak classifiers to improve predictive accuracy [34]. In this context, the algorithm utilizes three time–frequency features extracted from the motor current signal as machine health indicators, denoted as $f_1$, $f_2$, and $f_3$. AdaBoost works by iteratively training weak classifiers on a weighted dataset, where each weak classifier focuses on

specific aspects of the input features. These weak classifiers are combined in a weighted manner to form the ensemble classifier. Let $h_t(x)$ represent the prediction of the $t$-th weak classifier for the input feature vector $x$ and $\alpha_t$ represent its corresponding weight. The final prediction model, denoted as $H(x)$, is a function of the input feature vector $x$. It is computed by combining the predictions of all weak classifiers, weighted by their respective performances, as given in Equation (17):

$$H(x) = \sum_{t=1}^{T} \alpha_t \cdot h_t(x) \tag{17}$$

where $T$ is the total number of weak classifiers. AdaBoost with weak classifiers improves prediction by leveraging the strengths of multiple models that work well together. This makes it suitable for predicting the severity of faults in machine health monitoring. Weak classifiers with AdaBoost emerge as a superior technique for fault severity prediction due to several key advantages over alternative methods [34]. While Hidden Semi-Markov Models (HSMMs) offer flexibility in modeling variable-duration states, they may struggle with complex data patterns and require extensive parameter tuning [36]. Deep Learning Neural Networks, renowned for capturing intricate patterns, demand substantial computational resources and extensive data for effective training, limiting their practical applicability in resource-constrained environments [37]. Ensemble Learning Methods, including Random Forests and Gradient Boosting Machines, provide robustness against overfitting and enhance generalization by combining multiple models. However, they may suffer from increased computational complexity and a lack of interpretability [38]. Fuzzy Logic Systems offer a framework for handling uncertainty but may struggle with capturing complex nonlinear relationships in data [39]. In contrast, AdaBoost with weak classifiers achieves high predictive accuracy while maintaining computational efficiency [40]. By iteratively training weak learners on weighted subsets of data, AdaBoost focuses on misclassified samples, gradually improving performance. This approach effectively addresses the challenges of noisy data and complex data patterns, making it particularly suitable for fault severity prediction tasks in industrial settings [40]. Additionally, the simplicity of weak classifiers enhances model interpretability, facilitating insights into the underlying factors contributing to fault severity. It turns out that AdaBoost with weak classifiers is a strong and effective way to accurately predict the severity of a fault, which can be used in a number of real-world situations [40].

## 3. Method

In this research, data collection involved gathering vibration, current, and voltage signals from operational DC motors using high-fidelity sensors and acquisition systems, capturing signals under diverse conditions. After that, visual inspection and Fast Fourier Transform (FFT) spectral analysis helped find patterns that could tell the difference between healthy and faulty states. Logarithmic scaling made frequency domain visualization easier. Frequency analysis pinpointed fault-associated components within specific ranges, extracting features crucial for gear fault diagnosis. Leveraging machine learning, particularly the AdaBoost Ensemble Approach with weak classifiers, classifiers were trained on these features for fault diagnosis, while model parameters were meticulously calculated to optimize predictive performance. Testing the method on different sets of data and different operating conditions proved that it accurately found and predicted gear faults in DC motors, demonstrating that it can be used in real life.

### 3.1. Feature Extraction and Fault Diagnosis

The fault diagnosis and prognosis methodology for the DC motor's gear system involves a detailed analysis of features extracted from the motor's current signals. Various techniques for feature extraction, such as time-domain and frequency-domain analysis, are explored. This study's proposed methodology focuses on utilizing a boosting algorithm with weak classifiers for fault diagnosis and prognosis. During diagnosis, an ensemble

classifier is trained on the extracted features from the motor current signal. The ensemble classifier is adept at classifying the present fault state of the machine based on the extracted features. As for prognosis, the ensemble classifier is utilized to anticipate future fault severity conditions. By amalgamating predictions from multiple weak classifiers trained during the diagnosis phase, the boosting algorithm computes the most probable next state and assesses the severity of potential faults. The weights of the weak classifiers are determined during the training phase of the boosting algorithm, where classifiers exhibiting better performance are assigned higher weights. This meticulous weight allocation ensures that well-performing classifiers contribute more significantly to the ensemble's decision-making process. The proposed methodology offers a reliable and comprehensive approach to fault diagnosis and prognosis, leveraging ensemble learning techniques to provide accurate predictions of fault severity in machine health monitoring, thereby enhancing reliability and efficiency in industrial operations.

*3.2. Model Parameter Calculation*

As discussed previously, the proposed methodology leverages a boosting algorithm with weak classifiers to facilitate fault diagnosis and prognosis, eliminating the reliance on HMMs. The boosting algorithm determines crucial model parameters, including the weighting coefficients of the weak classifiers, during its training phase. In scenarios where extensive historical data are unavailable, alternative methods for model parameter calculation are explored. Extensive-scale testing involves evaluating significant samples of faulty machines, enabling the natural progression of using methods like finite element analysis, stress analysis, and nondestructive testing, while fatigue analysis looks at how the starter motor gear and flywheel wear over time. Additionally, online estimation estimates transition probabilities during vehicle operation, leveraging similarities to previous transitions. These approaches provide valuable insights into model parameter calculation, ensuring robust fault diagnosis and prognosis methodologies in machine health monitoring applications. However, practical constraints limit the feasibility of the first two options. Online estimation, the third method, is applicable only during machine operation, where faults develop naturally. For laboratory setups, heuristic methods are employed to estimate these probabilities. The proposed methodology presents a pragmatic approach to model parameter calculation, harnessing ensemble learning techniques for precise fault diagnosis and prognosis in machine health monitoring. The equations of the boosting algorithm elucidate the mathematical formulas utilized for training and prediction making with the ensemble classifier. This is very important for correctly identifying faults and predicting their outcomes in machine health monitoring. The weighted error rate $err_t$ shown in Equation (18) for the $t$-th weak classifier $h_t$ is computed by summing the weights $w_i$ of misclassified samples and dividing by the total sum of sample weights.

$$\text{err}_t = \frac{\sum_{i=1}^{N} w_i \cdot \text{indicator}(h_t(x_i) \neq y_i)}{\sum_{i=1}^{N} w_i} \tag{18}$$

The weight $\alpha_t$, as shown in Equation (19) assigned to the $t$-th weak classifier $h_t$, is determined based on the computed error rate $\text{err}_t$. It represents the contribution of $h_t$ to the final ensemble classifier.

$$\alpha_t = 0.5 \cdot \log\left(\frac{1 - \text{err}_t}{\text{err}_t}\right) \tag{19}$$

Equation (20) explains sample weights $w_i$ are updated based on the weak classifier $h_t$ performance, adjusting them to emphasize correctly classified samples and de-emphasize misclassified ones.

$$w_i = w_i \cdot \exp(-\alpha_t \cdot y_i \cdot h_t(x_i)), \quad i = 1 \text{ to } N \tag{20}$$

The sample weights $w_i$ are normalized to sum up to 1, as shown in Equation (21), maintaining their probabilistic interpretation:

$$w_i = \frac{w_i}{\sum_{i=1}^{N} w_i}, \quad i = 1 \text{ to } N \tag{21}$$

The ensemble classifier $H(x)$ can be calculated using Equation (22), aggregating the predictions of all weak classifiers $h_t$ weighted by their respective weights $\alpha_t$, and producing the final prediction. The model parameter calculation is crucial to the proposed methodology for fault diagnosis and prognosis in machine health monitoring:

$$H(x) = \text{sign}\left(\sum_{t=1}^{T} \alpha_t \cdot h_t(x)\right) \tag{22}$$

By exploring alternative methods such as large-scale testing, fatigue analysis, and online estimation, this methodology offers practical solutions for scenarios where extensive historical data may be unavailable. The detailed equations presented elucidate the training and prediction process of the boosting algorithm with weak classifiers, highlighting their significance in achieving accurate fault severity predictions. By leveraging insights from robust techniques for parameter calculation, the subsequent pseudocode defined in Algorithm 1 delineates the implementation of AdaBoost with weak classifiers for fault detection in DC motor gears. This algorithm is very good at finding patterns that point to gear problems by improving the classification model over and over again using weighted training data. Consequently, it facilitates the development of dependable predictive maintenance strategies for DC motor systems.

---

**Algorithm 1** AdaBoost with weak classifiers

---

1:  **Input:** Training dataset $D = \{(x_1, y_1), (x_2, y_2), ..., (x_n, y_n)\}$ where $x_i$ are feature vectors and $y_i \in \{-1, +1\}$ are labels, number of weak classifiers $T$
2:  **Output:** Final strong classifier $H(x)$
3:  **procedure** ADABOOST($D, T$)
4:      Initialize weights $w_i = \frac{1}{n}$ for $i = 1, 2, ..., n$
5:      Initialize an empty list $H$ to store weak classifiers
6:      **for** $t = 1$ to $T$ **do**
7:          Train a weak classifier $h_t(x)$ using weighted dataset $D_t = \{(x_i, y_i, w_i)\}_{i=1}^{n}$
8:          Calculate error $\varepsilon_t = \sum_{i=1}^{n} w_i \cdot \mathbb{K}(h_t(x_i) \neq y_i)$
9:          Ensure $\varepsilon_t < 0.5$ to ensure classifier performance above random guessing
10:         Compute weak classifier weight $\alpha_t = \frac{1}{2} \ln\left(\frac{1-\varepsilon_t}{\varepsilon_t}\right)$
11:         Update weights: $w_i \leftarrow w_i \cdot \exp(-\alpha_t y_i h_t(x_i))$ for $i = 1, 2, ..., n$
12:         Normalize weights: $w_i \leftarrow \frac{w_i}{\sum_{i=1}^{n} w_i}$ for $i = 1, 2, ..., n$
13:         Append weak classifier $h_t(x)$ to list $H$
14:     **end for**
15:     **return** Final strong classifier $H(x) = \text{sign}\left(\sum_{t=1}^{T} \alpha_t h_t(x)\right)$
16: **end procedure**

---

The AdaBoost algorithm with weak classifiers defined in Algorithm 1 operates by iteratively refining a solid classifier from a collection of weak learners. Initially, the algorithm assigns equal weights to all training samples, ensuring each observation contributes equally to the learning process. Through a series of iterations, known as rounds, weak classifiers are trained on weighted subsets of the training data. These weak classifiers are typically simple models, such as decision stumps, which focus on capturing specific patterns in the data. After training each weak classifier, its performance is evaluated by calculating the error rate, which measures the discrepancy between the predicted and actual labels. To maintain the classifier's performance above random guessing, the algorithm requires each weak classifier's error rate to remain below 0.5. The weight of each weak

classifier is then determined based on its error rate, with more accurate classifiers assigned higher weights. Additionally, the weights of the training samples are adjusted to prioritize misclassified instances, effectively emphasizing challenging data points in subsequent iterations. After normalization, the weak classifiers are combined into a final robust classifier using a weighted voting scheme, where each weak classifier's contribution is weighted based on its performance. This ensemble approach allows AdaBoost to effectively leverage the collective knowledge of the weak classifiers, resulting in a robust classifier capable of accurately predicting DC motor gear faults. Through its iterative learning process and adaptive weight adjustment, AdaBoost with weak classifiers enhances the reliability of predictive maintenance strategies for DC motor systems, facilitating early detection and mitigation of gear faults to ensure uninterrupted operation and prolong the lifespan of the equipment. This holistic approach underscores the importance of robust model parameter calculation techniques in developing effective machine health monitoring systems. The explanation effectively delineates the methodology's approach to determining vital model parameters for fault diagnosis and prognosis in machine health monitoring. It adeptly discusses alternative methods such as large-scale testing, fatigue analysis, and online estimation, showcasing the methodology's adaptability in scenarios where extensive historical data may be scarce. The detailed equations show how the boosting algorithm with weak classifiers is trained and how it makes predictions. This shows how important they are for getting accurate predictions of fault severity. This comprehensive explanation provides valuable insights into how robust model parameter calculation techniques contribute to the methodology's efficacy in developing dependable machine health monitoring systems.

## 4. Implementation

The DC motor under investigation is an integral part of automotive starting systems. Faults within the gear mechanism, such as gear wear, misalignment, or damage, can impede the functionality of the starter motor system by disrupting the smooth transmission of rotational energy from the motor to the engine's flywheel [41]. For instance, worn or damaged gear teeth may lead to improper engagement and ineffective engine starting [42]. Similarly, misalignment of the gear components can cause uneven wear and increased friction, reducing the efficiency of the starter motor system. These gear faults hinder engine ignition and can accelerate wear and tear on the motor components, necessitating timely diagnosis and repair to ensure optimal vehicle performance. Moreover, faults within the gear mechanism, particularly in the teeth, can have significant repercussions on the performance of the starter motor system [41]. When gear teeth are worn, damaged, or misaligned, the smooth transmission of rotational energy is compromised, leading to gear slippage, where the gear fails to engage with the flywheel properly [41,42]. Additionally, damaged or misaligned teeth can create uneven contact surfaces, increasing friction and wear on the gear components over time. These problems can escalate if left unaddressed, potentially causing further damage to the motor and associated components [41]. Thus, addressing gear tooth problems is crucial to maintaining the integrity of the starter motor system and ensuring reliable engine start-up [42]. Given the inherent complexities of starter motor operation, accurately determining its lifespan is challenging. However, the proposed method offers a solution by providing predictions for different fault severities that may occur during subsequent starting attempts and the likelihood of complete failure. This estimation of the failure state can indirectly indicate the motor's remaining useful life. Demonstrations were conducted using real-world data collected from laboratory experiments to validate the approach.

### 4.1. Experimental Setup

A 12V battery powered a comprehensive engine module that included the starter motor assembly in the laboratory configuration. Faults are intentionally induced in the motor gear mechanism to replicate real-world scenarios where gear damage occurs. This deliberate process involves controlled manipulation of the gear system to introduce spe-

cific types of faults, such as One-Half Tooth, One Tooth, Two Teeth, and Two Half Teeth. Techniques like controlled wear, deliberate misalignment, or simulated damage to gear teeth are employed to replicate common issues encountered in practice accurately. Each induced fault is meticulously created to closely mimic the characteristics and severity of the corresponding fault category. This meticulous process ensures that the induced faults accurately represent the types of damage observed in real-world situations, providing valuable insights into the behavior and performance of the motor gear mechanism under different fault conditions. The PC controlled the starter motor solenoid via signal conditioning devices. The data acquisition program initiated the generation of starting signals, while a commercially available data acquisition card (NI DAQ-6229) and Labview software were used for sampling. Signals such as motor current, battery voltage, and vibration were collected. However, this study focused solely on the motor current. A position sensor, equipped with an optical pulse counter, was utilized to acquire precise measurements of gear locations and motor speed. The diagram in Figure 1 provides an overview of the experimental setup.

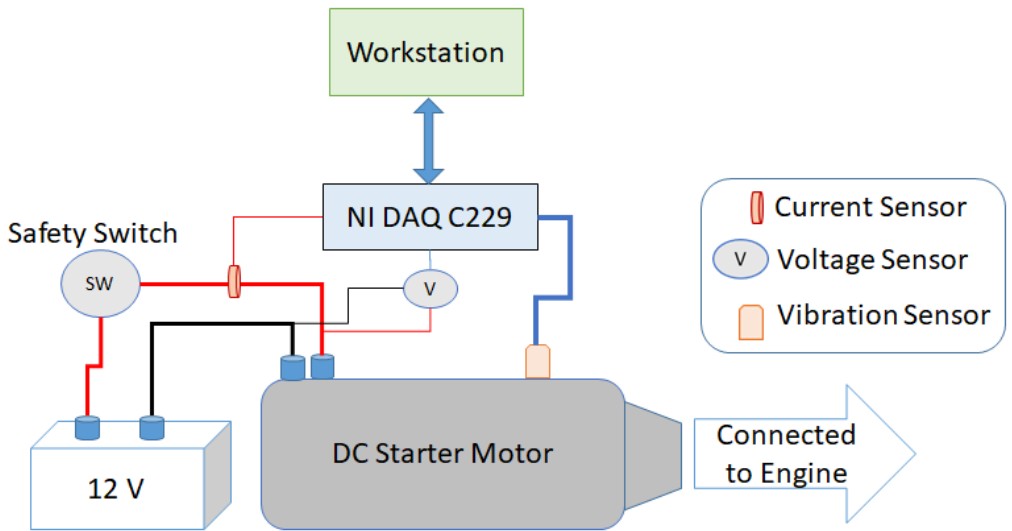

**Figure 1.** Block Diagram of Experimental Setup.

### 4.2. Operation of Motor and Nature of Fault

During each starting attempt, the starter DC motor operates for approximately 1 s. Within this time frame, the starter motor gear engages with the flywheel attached to the engine crankshaft, a critical step in the starting process. During the compression cycle of each cylinder, the torque exerted by the starter motor is significant, leading to increased force on the teeth of the starter gear. When the teeth are damaged, the impact and meshing pattern differ, causing variations in the motor current. This results in a periodic signal synchronized with the engine's compression and expansion frequencies. In faults, such as damaged teeth, high frequencies are superimposed on this signal due to knock. In an 8000 ms timeframe, which accommodates approximately eight starting attempts of 1 s each, the motor's operational cycles are observed, reflecting the dynamic engagement of the starter gear with the engine flywheel. The current signature is shown in Figure 2.

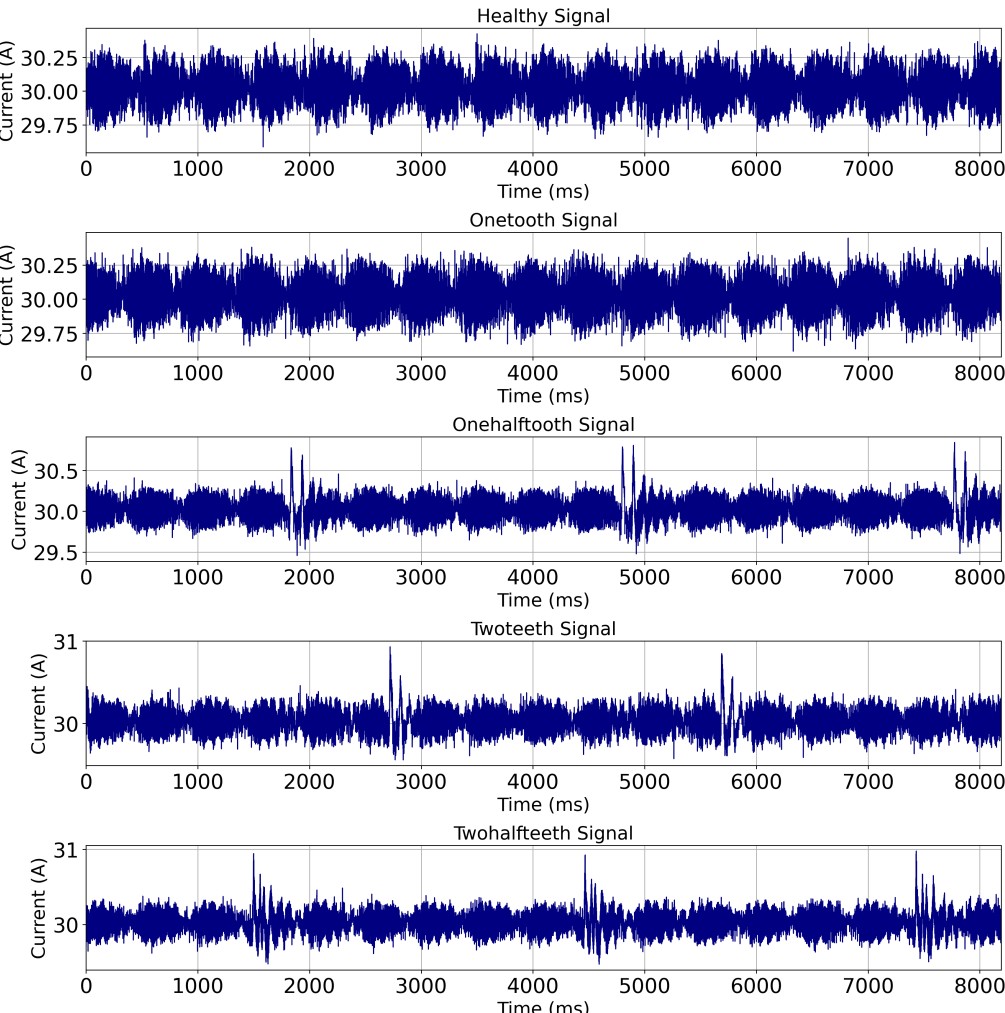

**Figure 2.** Current signatures of starter DC motor operation with healthy and faulty conditions.

### 4.3. Spectral Analysis

In the spectral analysis section, the vibration and current signals from the DC motor are put through a full frequency domain analysis to find fault signatures. The Fast Fourier Transform (FFT) algorithm is applied to convert the time-domain signals into frequency-domain representations, enabling the identification of specific frequency components associated with various fault conditions. By analyzing the amplitude and frequency characteristics within distinct frequency ranges, the spectral analysis provides valuable insights into the nature and severity of faults in the DC motor system. In addition, the spectral features that are taken from the vibration and current signals are used as inputs for later machine learning algorithms. This makes fault diagnosis and prediction more accurate. The Fast Fourier Transform (FFT) is utilized to convert the time-domain signals $x(t)$ into their corresponding frequency-domain [43] representations $X(f)$, defined in Equation (23):

$$X(f) = \int_{-\infty}^{\infty} x(t)e^{-j2\pi ft}dt \tag{23}$$

where $f$ represents frequency in Hz, $t$ denotes time in seconds, and $j$ denotes the imaginary unit. The FFT result is often plotted with logarithmic scaling to emphasize frequency components that might be hidden in linear plots. The logarithmic scaling of the FFT result $|X(f)|$ is given by Equation (24):

$$\mathrm{Log}_{10}|X(f)| = 10 \cdot \log_{10}(|X(f)|) \tag{24}$$

This transformation enhances the visualization of frequency components, particularly those with lower amplitudes, aiding in identifying fault-related signatures. The analysis is shown in Figure 3.

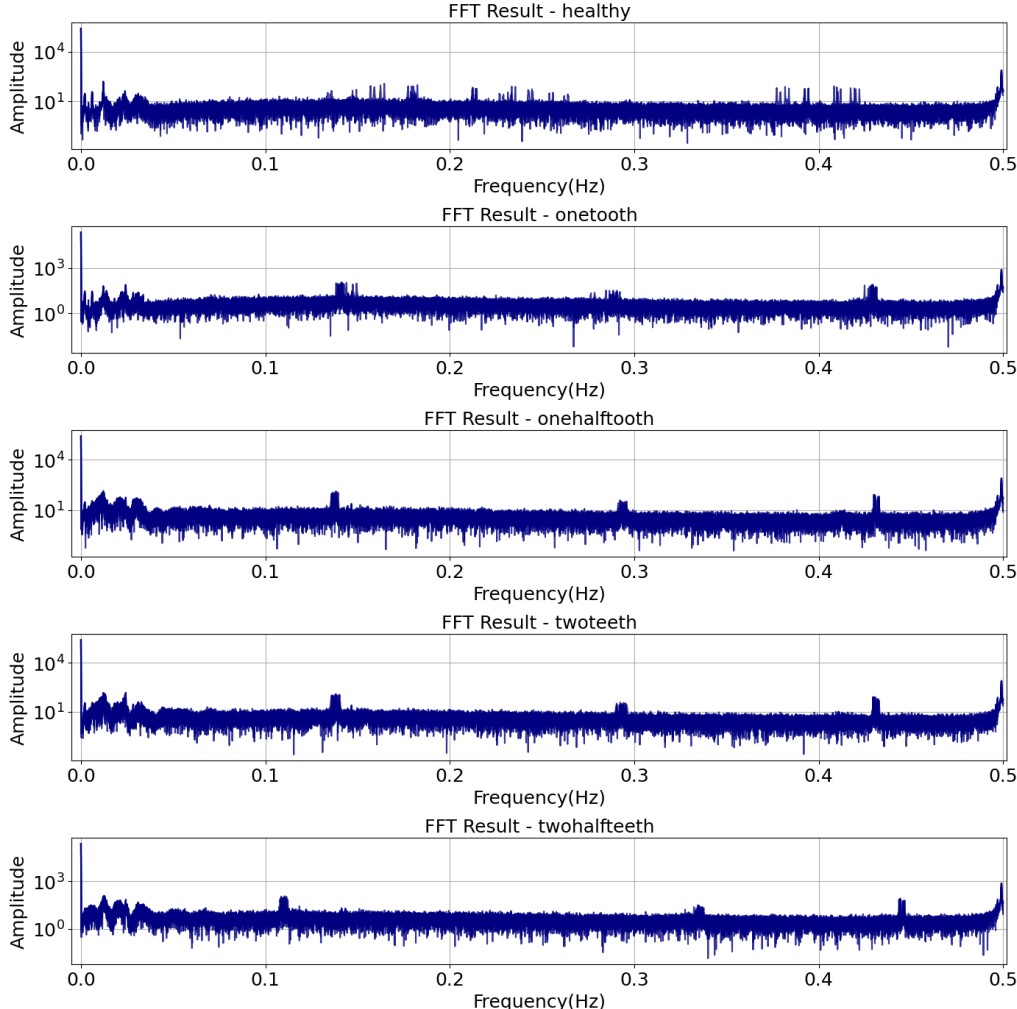

**Figure 3.** FFT analysis results with logarithmic scaling.

### 4.4. Frequency Range Analysis and Feature Extraction

In spectral analysis, distinct frequency ranges are meticulously examined to uncover fault-related signatures. This process involves dividing the frequency spectrum into specific bands, within which the amplitude and frequency of dominant peaks are meticulously analyzed. By focusing on these bands, fault-related frequencies can be isolated and their severity characterized. Critical spectral features, including peak amplitudes, peak frequencies, and spectral bandwidths, are then extracted from these frequency-domain representations. These features play a crucial role as discriminative parameters for machine learning algorithms, empowering them to achieve precise fault diagnosis and prognosis. Through this detailed spectral analysis, the methodology gains insights into the underlying fault mechanisms, enabling effective monitoring and maintenance strategies for the DC motor. Figure 4 illustrates the spectral features of a single sample within the frequency range of 0.1 to 0.2 Hz and 0.2 to 0.3 Hz, showcasing important fault-related parameters such as peak amplitudes and frequencies. In the frequency range of 0.1 to 0.2 Hz, an increase in the amplitude of the peaks correlates with a higher severity of faults. Conversely, in the frequency range of 0.2 to 0.3 Hz, a decrease in peak amplitude is associated with a more pronounced fault occurrence. These findings indicate a direct

relationship between fault severity and the strength of specific frequency components within each frequency range.

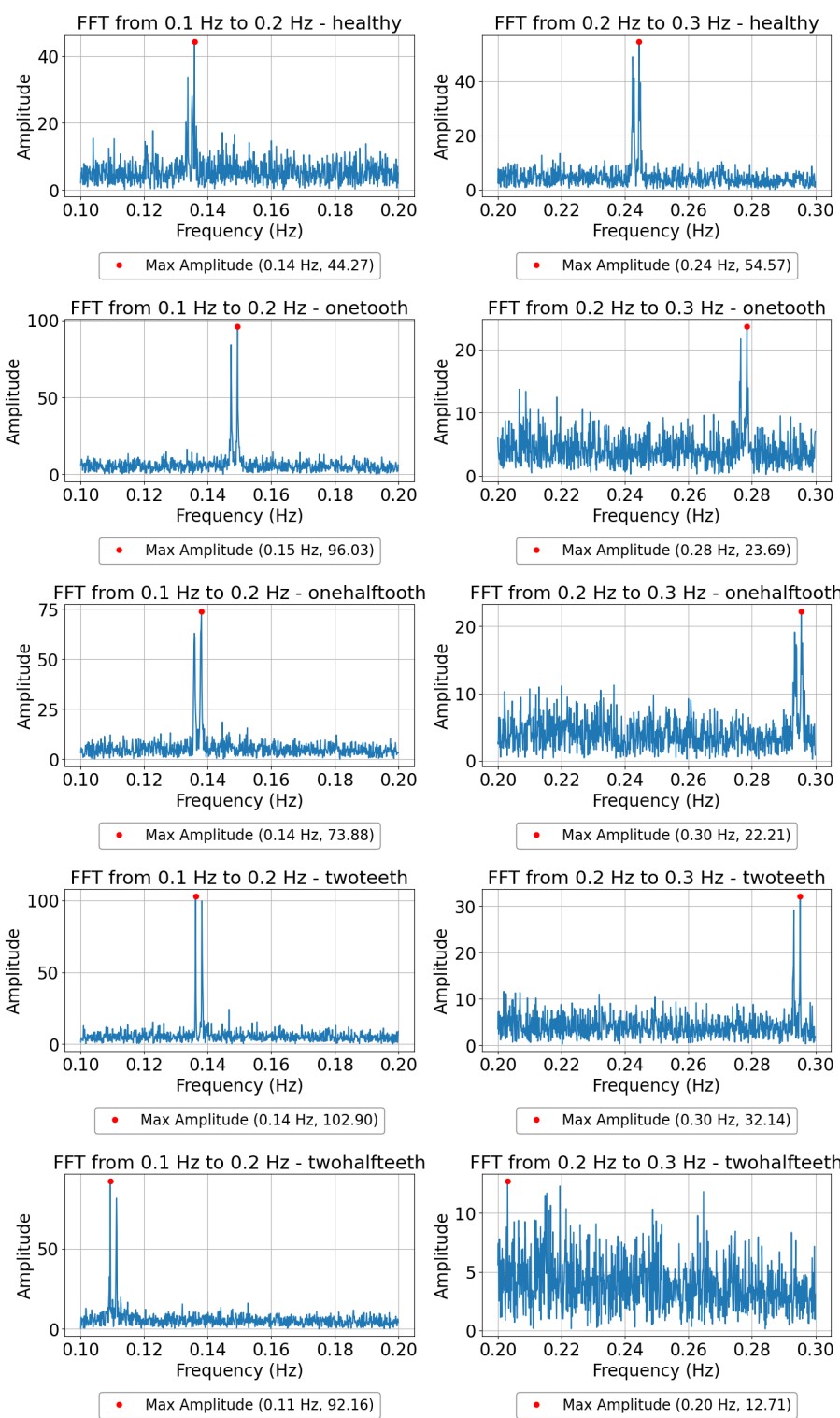

**Figure 4.** FFT plots showcasing the dominant frequency components within the 0.1 to 0.2 and 0.2 to 0.3 Hz ranges for different fault levels in DC motor operation.

### 4.5. Diagnosis Algorithm Implementation

In this study, the AdaBoost (Adaptive Boosting) algorithm was applied for fault classification using the provided dataset. AdaBoost is an ensemble learning technique

that combines multiple weak classifiers to form a robust classifier [40]. Let $X$ denote the feature matrix and $y$ denote the target variable, assumed to be 'Fault Status' in this context. The dataset was partitioned into training and testing sets, with a test size of 20% and a random state of 40 to ensure reproducibility. The weak classifier chosen for AdaBoost was a Decision Tree Classifier with a maximum depth of 1, denoted as $h(x; \theta_i)$, where $\theta_i$ represents the parameters of the weak classifier. The Decision Tree Classifier is defined in Equation (25).

$$h(x; \theta_i) = \begin{cases} 1 & \text{if } x_j < s \\ -1 & \text{otherwise} \end{cases} \tag{25}$$

$x_j$ represents the feature value, and $s$ is the splitting threshold determined during training. For classification tasks, a Decision Tree Classifier serves as a supervised learning algorithm. It works by recursively partitioning the feature space into regions and assigning a class label to each region. The Decision Tree consists of root, internal, and leaf nodes. Each internal node represents a decision based on the value of a feature, and each leaf node represents a class label. Let $\mathcal{X}$ denote the feature space, $\mathcal{Y}$ denote the set of class labels, and $D = \{(x_1, y_1), (x_2, y_2), \dots, (x_N, y_N)\}$ denote the dataset, where $x_i$ is a feature vector and $y_i$ is the corresponding class label. A Decision Tree Classifier recursively splits the feature space into regions using decision rules. Each tree node applies a decision rule with a splitting threshold $s$ to a feature $x_j$. The process takes the left branch if the feature value $x_j$ is less than the threshold $s$; otherwise, it takes the right branch. The process continues until it meets a stopping criterion, which could be reaching a maximum depth or a minimum number of samples in a node. Mathematically, a Decision Tree Classifier at each internal node $t$ applies a decision rule based on a feature $x_j$ and a threshold $s$. Let $R_t$ denote the region of feature space associated with node $t$. The decision rule can be represented as explained in Equation (26):

$$R_t = \begin{cases} \{x \in \mathcal{X} : x_j < s\} & \text{if } x_j < s \\ \{x \in \mathcal{X} : x_j \geq s\} & \text{otherwise} \end{cases} \tag{26}$$

The decision rule is determined during the training phase by selecting the feature $x_j$ and threshold $s$ that optimally split the data according to a specific criterion, such as minimizing impurity or maximizing information gain. Once the Decision Tree is constructed, class labels are assigned to each leaf node based on the majority of the training samples in that region. This study used a Decision Tree Classifier with a maximum depth of 1, meaning that the Decision Tree consists of only a root node and two leaf nodes. This simple Decision Tree serves as a weak learner within the AdaBoost Ensemble. In our implementation, a Decision Tree Classifier with a maximum depth of 1 was denoted as $\text{DT}(d = 1, R = 40)$. An AdaBoost classifier with base estimator $b = \text{DT}(d = 1, R = 40)$, number of estimators $n = 100$, and random state $R = 42$ was utilized. After initializing the Decision Tree Classifier as the weak learner, the AdaBoost algorithm is applied to sequentially fit the weak classifiers to the data and adjust the weights of misclassified samples to emphasize difficult-to-classify instances. The final robust classifier, denoted as $H(x)$ and explained in Equation (27), is a linear combination of the weak classifiers weighted by their performance:

$$H(x) = \text{sign}\left( \sum_{i=1}^{T} \alpha_i h(x; \theta_i) \right) \tag{27}$$

where $T$ is the total number of weak classifiers, $\alpha_i$ is the weight assigned to the $i$th weak classifier, and $\text{sign}(\cdot)$ is the sign function. The AdaBoost algorithm was trained on the training data $(X_{\text{train}}, y_{\text{train}})$ and evaluated on the testing data $(X_{\text{test}}, y_{\text{test}})$. The performance of the AdaBoost classifier was assessed using various metrics, such as accuracy.

### 4.6. Prognosis Algorithm Implementation

A prognosis becomes feasible if the features extracted from samples with increasing fault severity display a discernible trend. Faults of varying severity were induced in four machines by grinding one tooth. To identify such a trend, A clear pattern emerges by systematically analyzing the extracted features as fault severity increases, enabling the formulation of a prognosis model. This model becomes crucial for predicting and preventing potential machine failures, optimizing maintenance schedules, and ensuring uninterrupted operations. This proactive approach significantly enhances the efficiency and reliability of the machinery, resulting in improved productivity and cost savings. This prognostic endeavor necessitates an explanation of the algorithm, specifically focusing on AdaBoost, an Ensemble Learning Method for regression tasks. This method combines multiple weak regressors, particularly Decision Tree regressors, to create a robust regressor, enhancing prediction accuracy. The algorithm fits weak regressors to the data one step at a time, changing the weights of samples that were wrongly predicted to focus on the most difficult cases. The last regressor, which is a weighted sum of the weak regressors, gives us a full model for prognosis prediction that was trained and tested on different datasets. Mathematically, the Decision Tree regressor is defined in Equation (28).

$$h(x; \theta_i) = \begin{cases} y_1 & \text{if } x_j < s \\ y_2 & \text{otherwise} \end{cases} \tag{28}$$

where $x_j$ represents the feature value, $s$ is the splitting threshold determined during training, and $y_1$ and $y_2$ are the predicted output values. The final strong regressor, denoted as $H(x)$, is a weighted sum of the weak regressors, as explained in Equation (29):

$$H(x) = \sum_{i=1}^{T} \alpha_i h(x; \theta_i) \tag{29}$$

where $T$ is the total number of weak regressors, and $\alpha_i$ is the weight assigned to the $i$th weak regressor. The AdaBoost algorithm is trained on the training data $(X_{\text{train}}, y_{\text{train}})$ and evaluated on the testing data $(X_{\text{test}}, y_{\text{test}})$. Performance metrics such as mean squared error (MSE) are used to assess the effectiveness of the regressor for prognosis.

## 5. Results

This section explores various machine learning techniques employed in predictive methodology for diagnosing and predicting gear problems in DC motors, emphasizing the effectiveness of AdaBoost with weak classifiers. A lot of well-known algorithms are talked about, including K-Means clustering, linear regression, Random Forest Classifiers, Gradient Boosting Machines (GBMs), Extreme Gradient Boosting (XGBoost), Logistic Regression Classifiers, Support Vector Machines (SVMs) with linear kernels, and AdaBoost with weak classifiers. Each algorithm is examined regarding its underlying principles, applications, and effectiveness in fault detection and diagnosis tasks in diagnosing and predicting gear problems in DC motors. Among these algorithms, AdaBoost stands out for its ability to boost the performance of weak classifiers, making it particularly suitable for this predictive methodology, where accurate and reliable predictions are crucial for maintaining the health and functionality of DC motor gears.

### 5.1. K-Means Clustering

This research investigated the effectiveness of K-Means clustering for fault diagnosis in DC motors using frequency and amplitude data. Despite identifying five clusters, the visual inspection of the clustered data revealed significant overlap, indicating a lack of distinct separation between fault conditions. This observation suggests that K-Means clustering may need to be better suited for accurately categorizing faults in DC motors based solely on these features. Potential factors contributing to this limitation include the complexity

and variability of faults in DC motors and the inherent sensitivity of K-Means clustering to initialization and dataset dimensionality. As shown in Figure 5, the overlap among clusters is evident, reinforcing the challenges associated with using K-Means clustering for fault diagnosis in DC motors.

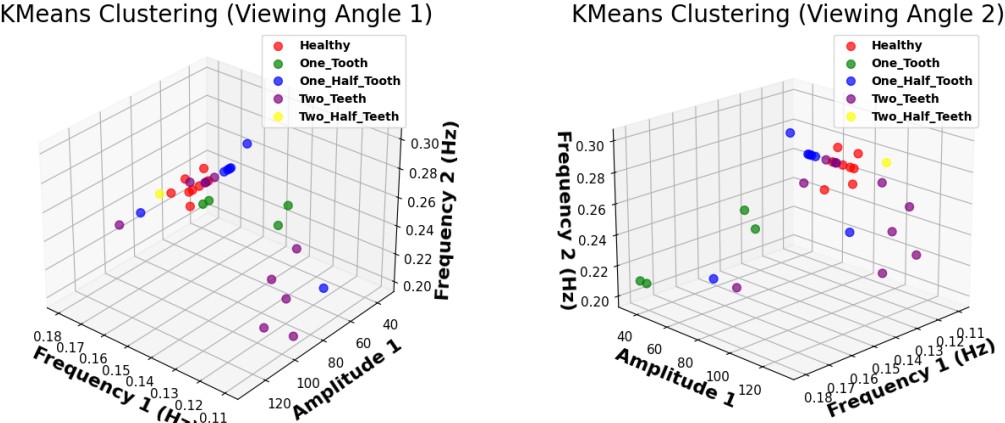

**Figure 5.** The scatter plot illustrates the clustering results obtained using K-Means algorithm for fault diagnosis in DC motors.

## 5.2. Linear Regression

It is known that linear regression is easy to understand and use. However, the use of this method to diagnose and predict gear problems in DC motors, as shown in the accompanying Figure 6, highlights the difficulties that come from not having enough data and the dataset's inherent limits. The linear regression model has a hard time capturing the complex relationships in the time–frequency features that are taken from motor current signals. This is especially true when the historical records are limited. This constraint significantly challenges the model's ability to generalize effectively and accurately predict fault conditions and remaining useful life. Because of this, linear regression is still a popular method, but it has trouble dealing with complex and nuanced data, which is made worse by a lack of data. This makes it less likely to be useful for reliable fault diagnosis and prognostics in DC motors. Future research efforts may need to explore alternative machine learning approaches to better accommodate the limited data availability and address the complexities inherent in fault diagnosis and prognostic tasks.

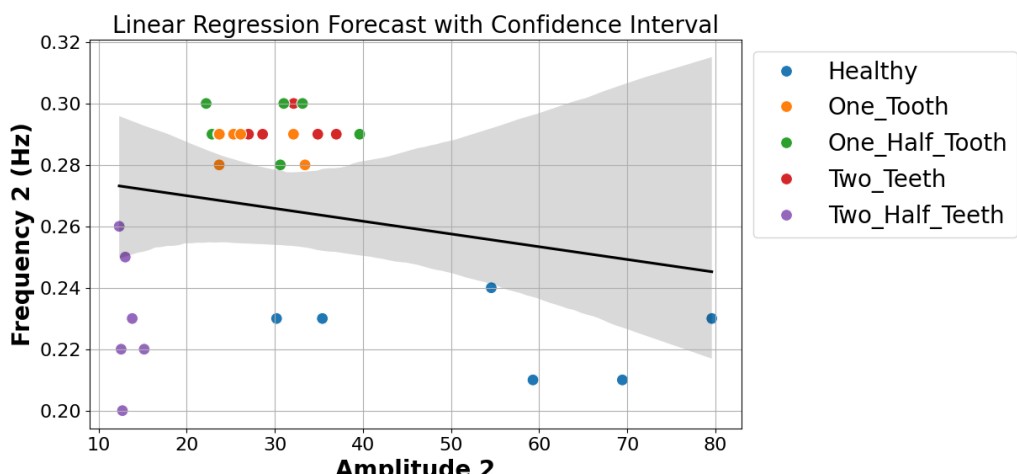

**Figure 6.** Scatter plot showing the relationship between Amplitude2 and Frequency2 with different fault statuses.

### 5.3. Long Short-Term Memory

When dealing with small datasets, particularly gear problems in DC motors, LSTM models encounter several challenges. Firstly, due to their large number of parameters, LSTMs are prone to overfitting, potentially memorizing training examples rather than generalizing well to unseen data. This challenge is especially pertinent in applications like diagnosing gear problems in DC motors, where the limited dataset may need to capture real-world conditions' total variability adequately. Consequently, the model's ability to generalize to unseen instances may need to be improved, leading to suboptimal performance. Additionally, the capacity of LSTMs to learn complex patterns is constrained by the scarcity of training examples, further exacerbating the risk of overfitting and hindering the model's ability to capture the underlying dynamics of gear faults. Moreover, small datasets often contain higher noise levels relative to the signal, amplifying the challenge of separating meaningful patterns from noise during training. Tuning hyperparameters also becomes challenging in this scenario, as there are limited data to estimate their impact on performance reliably. Finally, the risk of data leakage is heightened with small datasets, as inadequate partitioning between training and validation sets may lead to overly optimistic performance estimates. While the challenges mentioned are significant, the performance of the LSTM model can be visually evaluated through the generated plots, showcasing the predicted values compared with the actual values over time, as shown in the generated graphs in Figure 7. Here, 75% of the data are used for training and 25% are reserved for testing purposes.

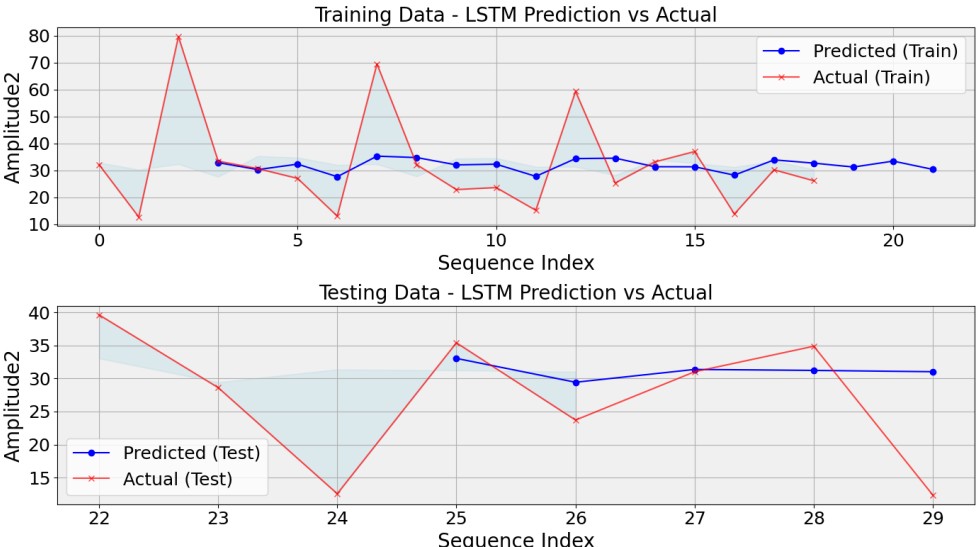

**Figure 7.** Comparison of LSTM predictions and actual values for training and testing data in gear fault diagnosis for DC motors.

Drawing inspiration from challenges in aviation prognostics [19], this study evaluates LSTM autoencoders for prediction, demonstrating their potential to enhance prognostic capabilities despite the limited availability of labeled data. However, LSTM autoencoders encounter significant hurdles, particularly in scenarios involving small datasets such as DC motor gear problems. Their intricate architectures, laden with parameters, increase the risk of overfitting, potentially impeding generalization. This challenge is pivotal in gear problem diagnosis, where limited datasets may inadequately capture real-world conditions, affecting performance. Furthermore, the scarcity of training examples restricts the autoencoder's ability to discern fault patterns. Small datasets often harbor elevated noise levels, complicating pattern extraction during training. Additionally, hyperparameter tuning becomes challenging due to insufficient data for accurate estimation. Improper partitioning between training and validation sets heightens the risk of data leakage, resulting in overly optimistic performance estimates. Despite these obstacles, visual inspection of

plots remains crucial for evaluation, facilitating comparison of predicted values over time and displaying predicted values alongside their corresponding actual values over time, as depicted in the graphs generated in Figure 8.

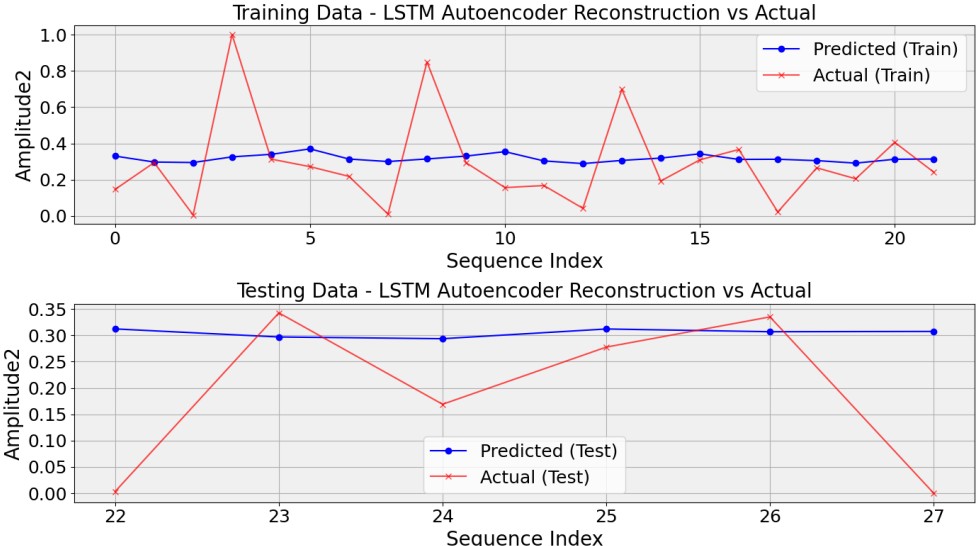

**Figure 8.** Evaluating LSTM autoencoder predictions against actual values in DC motor gear fault diagnosis.

### 5.4. Levenberg–Marquardt Neural Network

The methodology applied for detecting and managing faults in DC motor teeth, utilizing the Levenberg–Marquardt Neural Network (LMNN) model, draws parallels with an approach proposed for extending the longevity of power conversion systems [44]. This method integrates the Levenberg–Marquardt Back Propagation Neural Network (LM-BPNN) for state estimation and power routing, focusing on predictive maintenance in industrial contexts [44]. Specifically addressing fault detection, the LMNN model analyzes motor performance and fault data to discern patterns indicative of tooth faults within the gear system. Employing LMNN, a variant of the Backpropagation Neural Network (BPNN), trained via the Levenberg–Marquardt optimization algorithm, the architecture encompasses input, hidden, and output layers initialized with random weights. Through Levenberg–Marquardt (LM) algorithm-based training on provided data, including features derived from frequency and amplitude measurements alongside corresponding fault statuses, the model strives to optimize weights. However, limitations stemming from the small dataset or fault complexity may impede accurate fault detection and classification, as depicted in Figure 9. The restricted dataset size constrains the LMNN model's ability to capture the variability of fault manifestations and motor performance patterns, thereby limiting its performance. Despite these constraints, this approach outperforms LSTM and LSTM autoencoder-based techniques. Regarding Root Mean Square Error (RMSE), the LMNN method surpasses LSTM and LSTM autoencoder techniques. While the RMSE values for LSTM and LSTM autoencoder are 17 and 21, the LMNN technique achieves a significantly lower RMSE of 12. This disparity in RMSE values underscores the LMNN model's superior predictive accuracy compared with the LSTM and LSTM autoencoder models. Further refinement of the dataset and enhancements to the model's architecture and methodology are crucial to enhancing its effectiveness and reliability in real-world fault detection scenarios for DC motors used in starter applications. To address these challenges, practitioners may explore techniques such as data augmentation, transfer learning, regularization, leveraging pretrained models, and considering simpler model architectures or alternative approaches tailored to the specific constraints of small datasets in gear fault diagnosis for DC motors.

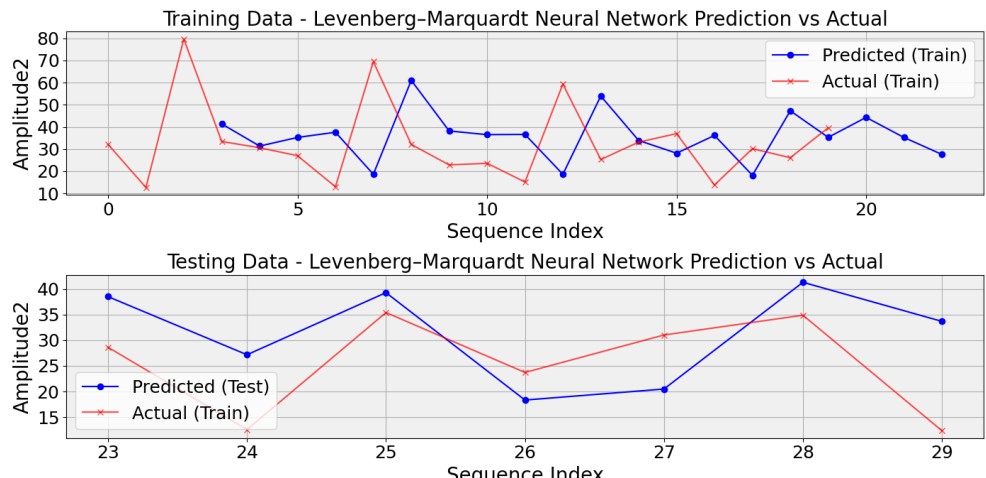

**Figure 9.** Evaluating Levenberg–Marquardt Neural Network predictions against actual values in DC motor gear fault diagnosis.

### 5.5. Random Forest Classifier

The Random Forest Classifier demonstrates mixed performance in classifying gear problems in DC motors, as reflected in its precision, recall, and F1-score metrics. While achieving perfect scores for the 'Healthy' and 'Two Half Teeth' classes, with precision and recall of 1.00, the classifier struggles notably with the 'One-Half Tooth' and 'One Tooth' categories, where it shows significant deficiencies, yielding precision, recall, and F1-score metrics close to zero. Overall, the classifier's accuracy of 50% indicates a notable misclassification rate across the dataset, as shown in Table 1. These discrepancies underscore challenges in accurately identifying specific gear fault categories, potentially stemming from factors such as imbalanced class distributions or limitations in the model's capacity to discern subtle patterns within the data. Addressing these weaknesses may necessitate adjustments in feature selection and model hyperparameters or exploring alternative classification algorithms better suited to handle the intricacies of gear fault diagnosis in DC motors.

**Table 1.** Performance Metrics of Random Forest Classifier.

| Class | Precision | Recall | F1-Score | Support |
|---|---|---|---|---|
| Healthy | 1.00 | 1.00 | 1.00 | 1 |
| One Half Tooth | 0.00 | 0.00 | 0.00 | 3 |
| One Tooth | 0.50 | 0.50 | 0.50 | 2 |
| Two Half Teeth | 1.00 | 1.00 | 1.00 | 2 |
| Two Teeth | 0.00 | 0.00 | 0.00 | 0 |
| **Accuracy** | | 0.50 | | |
| **Macro Avg** | | 0.50 0.50 0.50 8 | | |
| **Weighted Avg** | | 0.50 0.50 0.50 8 | | |

Table 1 presents the performance metrics of a Random Forest Classifier for classifying different categories or classes, including precision, recall, and F1-score, as well as overall accuracy, macroaverage, and weighted average values. Precision indicates the accuracy of optimistic predictions, measuring the proportion of correctly predicted positive instances among all instances predicted as positive. Recall, also known as sensitivity, gauges the ability of the classifier to capture all positive instances, calculating the ratio of correctly predicted positive instances to the total actual positives in the dataset. The F1-score, as the harmonic mean of precision and recall, offers a balanced measure of a classifier's performance, accommodating scenarios where either precision or recall is disproportionately low. Additionally, support values indicate the number of occurrences of each class in the dataset,

providing context for the precision and recall metrics. Overall accuracy reflects the ratio of correctly predicted instances to the total number of instances, while macro- and weighted averages offer comprehensive assessments across all classes, considering potential class imbalances. These metrics collectively offer comprehensive insights into the performance of a classifier across different classes and are vital for evaluating its effectiveness in classification tasks. This table summarizes the classifier's performance across various metrics, facilitating evaluation and comparison of its effectiveness in classifying different classes. Additionally, it should be noted that 80% of the data are used for training purposes, while the remaining 20% are used for testing. The same distribution is applied to the rest of the testing in this research. To ensure consistency and facilitate comparability in the evaluation process, we employ the same table format for all subsequent tests.

### 5.6. Gradient Boosting Classifier

The Gradient Boosting Classifier's performance metrics demonstrate comparable weaknesses to those observed in the Random Forest Classifier. Although achieving perfect precision and recall scores for the 'Healthy' and 'Two Half Teeth' classes, it notably struggles with the 'One-Half Tooth' and 'One Tooth' categories, exhibiting poor performance with precision, recall, and F1-score metrics approaching zero. Consequently, the classifier's overall accuracy is only 50%, indicating a significant misclassification rate across the dataset.

### 5.7. Extreme Gradient Boosting (XGBoost)

XGBoost, short for Extreme Gradient Boosting, is a sophisticated machine learning algorithm highly regarded for its exceptional performance across diverse classification and regression tasks. It operates on Gradient Boosting, sequentially training weak learners, often Decision Trees, to refine predictions and improve accuracy. The Classification Report for the XGBoost Classifier reveals a mixed performance across different classes. While achieving perfect precision and recall for the 'Healthy' and 'Two Half Teeth' categories, the classifier encounters challenges with the 'One-Half Tooth' and 'One Tooth' classes. Despite achieving perfect precision for 'One-Half Tooth,' its recall is notably lower, leading to a lower F1-score. Similarly, 'One Tooth' exhibits moderate precision and recall, resulting in an average F1-score. The absence of support instances for 'Two Teeth' results in zero precision and recall. With an overall accuracy of 62%, the classifier displays a moderate level of misclassification across the dataset, as shown in Table 2. The macroaverage F1-score of 0.60 underscores the classifier's moderate performance in generalizing across all classes. In contrast, the weighted average F1-score of 0.69 highlights its overall performance, giving higher weight to classes with more significant support. These insights suggest opportunities for refining the model to enhance classification accuracy and generalization capability.

**Table 2.** Performance Metrics of XGBoost Classifier.

| Class | Precision | Recall | F1-Score | Support |
|---|---|---|---|---|
| Healthy | 1.00 | 1.00 | 1.00 | 1 |
| One-Half Tooth | 1.00 | 0.33 | 0.50 | 3 |
| One Tooth | 0.50 | 0.50 | 0.50 | 2 |
| Two Half Teeth | 1.00 | 1.00 | 1.00 | 2 |
| Two Teeth | 0.00 | 0.00 | 0.00 | 0 |
| **Accuracy** | | 0.62 | | |
| **Macro Avg** | | 0.70 0.57 0.60 8 | | |
| **Weighted Avg** | | 0.88 0.62 0.69 8 | | |

### 5.8. AdaBoost Classifier with Logistic Regression Weak Learner

The AdaBoost Classifier with Logistic Regression Weak Learner achieves an accuracy of 62%, indicating its overall effectiveness in classifying instances across all categories as represented in Table 3. However, its performance varies significantly across different classes. While it demonstrates high precision and recall for the 'Healthy' and 'Two Half

Teeth' classes, it struggles with the 'One Tooth' class, exhibiting low precision, recall, and F1-score. This discrepancy suggests that while the classifier is generally successful in identifying specific categories, it encounters challenges in accurately classifying instances belonging to others, thereby impacting its overall performance.

**Table 3.** Performance Metrics of AdaBoost Classifier with Logistic Regression Weak Learner.

| Class | Precision | Recall | F1-Score | Support |
|---|---|---|---|---|
| Healthy | 0.67 | 1.00 | 0.80 | 2 |
| One-Half Tooth | 0.50 | 0.50 | 0.50 | 2 |
| One Tooth | 0.00 | 0.00 | 0.00 | 1 |
| Two Half Teeth | 1.00 | 0.67 | 0.80 | 3 |
| Two Teeth | 0.00 | 0.00 | 0.00 | 0 |
| **Accuracy** | | 0.62 | | |
| **Macro Avg** | | 0.43 0.43 0.42 8 | | |
| **Weighted Avg** | | 0.67 0.62 0.62 8 | | |

### 5.9. AdaBoost Classifier with SVM Weak Learner

The AdaBoost classifier with SVM (linear kernel) demonstrates poor performance across all classes, with precision, recall, and F1-score values of 0.00 for most classes. This indicates that the classifier needs help to classify instances belonging to each class correctly. The accuracy of 0.12 reflects the classifier's overall low performance in correctly predicting the class labels. The macro-average and weighted-average F1-scores are both 0.20, as explained in Table 4, suggesting a lack of effectiveness in capturing the actual positive rate and overall performance across all classes. The linear SVM classifier's inability to effectively separate the classes in the given dataset, leading to inadequate predictive capabilities, may be the cause of this poor performance.

**Table 4.** Performance Metrics of AdaBoost Classifier with SVM (linear kernel).

| Class | Precision | Recall | F1-Score | Support |
|---|---|---|---|---|
| Healthy | 1.00 | 1.00 | 1.00 | 1 |
| One-Half Tooth | 0.00 | 0.00 | 0.00 | 3 |
| One Tooth | 0.00 | 0.00 | 0.00 | 2 |
| Two Half Teeth | 0.00 | 0.00 | 0.00 | 2 |
| Two Teeth | 0.00 | 0.00 | 0.00 | 0 |
| **Accuracy** | | 0.12 | | |
| **Macro Avg** | | 0.20 0.20 0.20 8 | | |
| **Weighted Avg** | | 0.12 0.12 0.12 8 | | |

### 5.10. AdaBoost Ensemble with Decision Tree as Weak Classifiers

The AdaBoost Classifier with Decision Tree as a weak classifier does a great job overall, especially when it comes to correctly putting people into the 'Healthy' and 'Two Half Teeth' classes, where it obtained perfect precision, recall, and F1-scores. However, it struggled with the 'One-Half Tooth' class, achieving perfect precision but relatively lower recall and F1-score due to misclassifications. Notably, it failed to classify any instances correctly in the 'Two Teeth' class, resulting in zero precision, recall, and F1-score. Despite these shortcomings in certain classes, the classifier's weighted average F1-score of 0.92 indicates good overall performance across all classes. The AdaBoost Classifier with Decision Tree as a weak classifier achieved an overall accuracy of 0.88, reflecting its ability to classify 88% of instances correctly. Details are shown in Table 5. This high accuracy underscores the model's capacity to learn and generalize patterns effectively, leading to reliable predictions on unseen data. However, to comprehensively evaluate its performance, it is essential to analyze other metrics like precision, recall, and F1-score, providing insights into its performance across different classes and highlighting areas for refinement and optimization.

**Table 5.** Performance Metrics of AdaBoost Classifier with Decision Tree as Weak Classifier.

| Class | Precision | Recall | F1-Score | Support |
|---|---|---|---|---|
| Healthy | 1.00 | 1.00 | 1.00 | 2 |
| One-Half Tooth | 1.00 | 0.50 | 0.67 | 2 |
| One Tooth | 1.00 | 1.00 | 1.00 | 1 |
| Two Half Teeth | 1.00 | 1.00 | 1.00 | 3 |
| Two Teeth | 0.00 | 0.00 | 0.00 | 0 |
| **Accuracy** | | 0.88 | | |
| **Macro Avg** | | 0.80 0.70 0.73 8 | | |
| **Weighted Avg** | | 1.00 0.88 0.92 8 | | |

*5.11. Diagnosis and Prognosis of Fault*

The diagnostic outcomes underscore the effectiveness of the AdaBoost ensemble utilizing Decision Trees as weak classifiers in accurately identifying fault statuses. With an impressive 88% overall accuracy, the model exhibits notable precision, recall, and F1-scores across diverse fault categories, particularly excelling in identifying 'Healthy' and 'Two Half Teeth' cases. However, it faces challenges in accurately classifying 'One-Half Tooth' instances, displaying perfect precision but relatively lower recall, leading to a reduced F1-score. Notably, the model struggles with the 'Two Teeth' category, failing to make accurate predictions. Nevertheless, the weighted average F1-score of 0.92 indicates robust overall performance. Figure 10 showcasing the classification performance of the AdaBoost ensemble model using a confusion matrix. Each cell represents the number of samples classified into a particular fault status, aiding in classification accuracy assessment. The right subplot illustrates the feature's importance, indicating the contribution of each feature in predicting fault statuses. This comprehensive analysis provides insights into the model's diagnostic efficacy and the significance of different features in fault diagnosis.

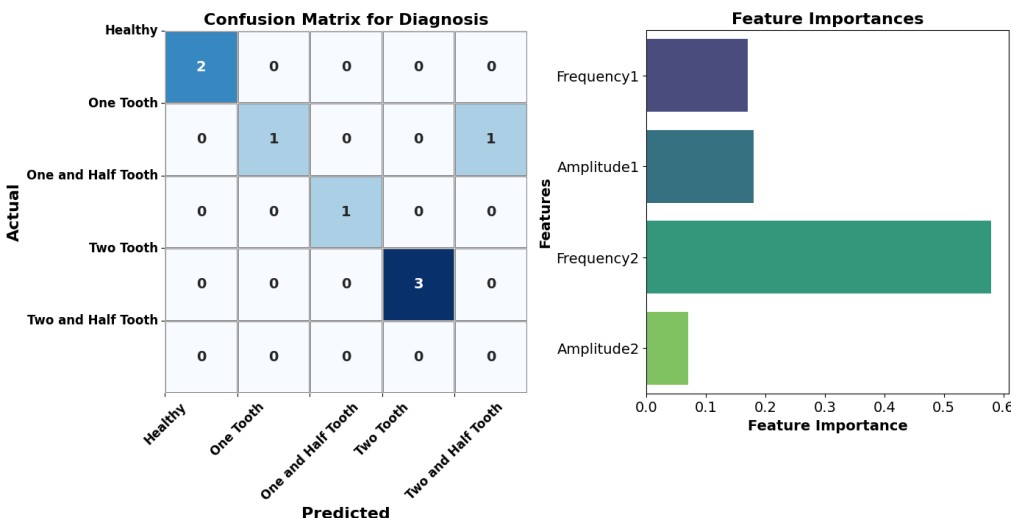

**Figure 10.** Diagnostic performance and feature importance analysis.

The mean squared error (MSE) of 1.12 shows how well the AdaBoost ensemble can guess the system's remaining useful life (RUL). Lower MSE values mean that the predicted and actual RUL values are more closely related. Overall, the AdaBoost ensemble with Decision Trees as weak classifiers emerges as a promising technique for fault diagnosis and prognosis, offering accurate insights into system health and future performance.

Figure 11 shows the relationship between the actual remaining useful life (RUL) values and the predicted RUL values generated by the AdaBoost regressor for prognosis. Each blue point represents an individual sample from the test dataset, where the x-coordinate corresponds to the actual RUL and the y-coordinate represents the predicted RUL. The

red dashed line indicates the line of perfect prediction, where the predicted RUL perfectly matches the actual RUL. Additionally, a green line, added as a line of best fit, illustrates the trend of the predicted RUL values concerning the actual RUL values. This line provides insights into the overall accuracy of the predictions and the direction of any systematic deviations between predicted and actual values. Including the green line in the legend enhances the graph's clarity by explicitly stating its purpose.

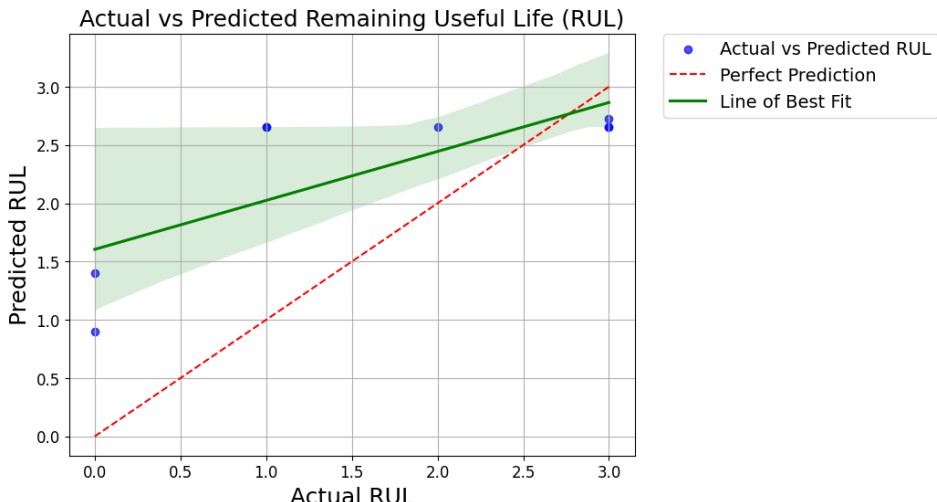

**Figure 11.** Performance evaluation of RUL prognosis through actual vs predicted scatter plot.

## 6. Discussion

This study delves into applying various machine-learning techniques for diagnosing and predicting gear problems in DC motors, focusing on the efficacy of AdaBoost with weak classifiers. Some well-known algorithms that have been studied are K-Means clustering, linear regression, Random Forest Classifiers, Gradient Boosting Machines (GBMs), Extreme Gradient Boosting (XGBoost), Logistic Regression Classifiers, Support Vector Machines (SVMs) with linear kernels, and AdaBoost with weak classifiers. Each algorithm's performance in fault detection and diagnosis tasks is scrutinized in the context of diagnosing and predicting gear problems in DC motors.

### 6.1. K-Means Clustering

The effectiveness of K-Means clustering in fault diagnosis was assessed using frequency and amplitude data. Although the algorithm identified five clusters, significant overlap was observed, suggesting inadequate separation between fault conditions. This limitation indicates that K-Means clustering may not be suitable for accurately categorizing faults in DC motors based solely on these features due to fault complexity and dataset dimensionality.

### 6.2. Linear Regression

Despite its simplicity, Linear regression struggled to capture nuanced relationships in time-frequency features extracted from motor current signals, mainly due to limited data availability and dataset constraints. Challenges in generalization were highlighted, raising concerns about its suitability for robust fault diagnosis and prognostics in DC motors.

### 6.3. Long Short-Term Memory (LSTM)

LSTM models faced challenges with small datasets, were prone to overfitting, and needed more training examples to generalize effectively. Additionally, we encountered difficulties with noise amplification and hyperparameter tuning, underscoring the need for alternative approaches tailored to the constraints of small datasets. Using LSTM autoencoder techniques also presented similar, if not exacerbated, challenges. Despite their

potential for capturing temporal dependencies and extracting meaningful features, these models encountered issues with small datasets, exacerbating overfitting tendencies and hindering generalization.

### 6.4. Levenberg–Marquardt Neural Network

The selection of the LMNN model for fault detection in DC motor teeth necessitates recognition of its inherent limitations. The quality and quantity of the training data heavily influence the model's success, and restricting the datasets can lead to erroneous fault identification. Furthermore, the intricate nature of fault patterns and fluctuations in motor function may pose a challenge to the model's efficacy. It may not always guarantee the convergence of the training process to optimal solutions, thereby requiring a significant amount of tuning. Given the aforementioned constraints, it may be necessary to investigate other algorithms or hybrid methodologies to enhance the effectiveness of fault detection.

### 6.5. Random Forest Classifier

The Random Forest Classifier exhibited mixed performance, excelling in certain fault categories while struggling with others. We highlighted challenges in accurately identifying certain gear fault categories, potentially stemming from imbalanced class distributions or model limitations in discerning subtle patterns.

### 6.6. Gradient Boosting Classifier

Similar to Random Forest, the Gradient Boosting Classifier faced challenges in accurately classifying certain fault categories, indicating limitations in identifying subtle patterns within the data.

### 6.7. Extreme Gradient Boosting (XGBoost)

XGBoost showed comparable weaknesses in accurately classifying certain fault categories, suggesting opportunities for refining the model to enhance classification accuracy and generalization capability.

### 6.8. AdaBoost Classifier with Logistic Regression and SVM Weak Learners

The AdaBoost classifiers with Logistic Regression and SVM (linear kernel) weak learners had varying levels of performance. They had major problems correctly classifying instances across all fault categories, which shows that they have trouble telling fault classes apart.

### 6.9. AdaBoost Ensemble with Decision Tree as Weak Classifiers

The AdaBoost Ensemble with Decision Tree as a weak classifier demonstrated strong overall performance, accurately classifying most fault categories. It showcased notable strengths in accurately identifying fault statuses, particularly excelling in certain fault categories. Despite observing some challenges in accurately classifying specific fault instances, the ensemble's overall performance remained robust.

### 6.10. Diagnostic and Prognostic Analysis

The diagnostic outcomes highlighted the AdaBoost ensemble's effectiveness in accurately identifying fault statuses, underscoring its potential for fault diagnosis in DC motors. Despite encountering challenges in accurately classifying certain fault instances, the ensemble maintained a high level of performance across various fault categories. The AdaBoost ensemble performed well in prognosis, indicating its potential utility for prognostic tasks related to DC motors.

Overall, the AdaBoost ensemble with Decision Tree as a weak classifier shows promise in fault diagnosis and prognosis for DC motors. While certain challenges exist, such as accurately classifying certain fault instances, the ensemble's overall performance and

potential for further refinement make it a valuable tool for addressing diagnostic and prognostic tasks in DC motors.

## 7. Conclusions

In conclusion, this analysis underscores the robust performance of the AdaBoost ensemble with Decision Trees as weak classifiers in diagnosing and predicting gear issues in DC motors. Despite the challenge of a relatively small dataset, AdaBoost demonstrates significant strengths in accurately identifying fault statuses. Achieving an impressive 88% overall accuracy, AdaBoost excels particularly in categorizing instances such as 'Healthy' and 'Two Half Teeth'. While encountering some limitations with certain fault categories, AdaBoost maintains a high level of performance with a weighted average F1-score of 0.92. These findings highlight the effectiveness of AdaBoost with Decision Trees in fault diagnosis, providing valuable insights into system health despite the constraints of limited data availability. With a mean squared error (MSE) of 1.12, this ensemble also does a good job of predicting the system's remaining useful life (RUL). This shows that it can make predictions.Overall, this study shows that AdaBoost with Decision Trees is very good at finding and predicting gear problems in DC motors. However, it is important to note that these results are only valid for the specific experiment setup and dataset used in this manuscript. Therefore, this warrants further validation on diverse datasets and real-world applications to ascertain the broader applicability and generalizability of AdaBoost with Decision Trees, despite its promising potential for fault diagnosis and prognosis in DC motor gear systems.

**Author Contributions:** S.S.H.Z. spearheaded the project, while S.S.H. and S.S.H.Z. collaborated on algorithm design. S.S.H. implemented the algorithms using an open-source platform, conducted simulations, and validated the findings in a laboratory setting under S.S.H.Z.'s guidance. The manuscript was authored by S.S.H. with oversight from S.S.H.Z., and all authors contributed to its review and approval. All authors have read and agreed to the published version of the manuscript.

**Funding:** Regarding funding, this research was self-funded and did not receive external financial support.

**Institutional Review Board Statement:** Not applicable.

**Informed Consent Statement:** Not applicable.

**Data Availability Statement:** The data are included in this article.

**Conflicts of Interest:** The authors declare no conflicts of interest.

## Abbreviations

The following abbreviations are used in this manuscript:

| | |
|---|---|
| DC | Direct Current |
| LSTM | Long Short-Term Memory Network |
| LMNN | Levenberg–Marquardt Neural Network |
| RUL | Remaining Useful Life |
| TDDB | Time-Dependent Dielectric Breakdown |
| HMM | Hidden Markov Model |
| MHGPAN | MultiHop Graph Pooling Adversarial Network |
| DFL | Distributed Federated Learning |
| WM-CVAE | Weighted Modified Conditional Variational Auto-Encoder |
| GBM | Gradient Boosting |
| XGBoost | Extreme Gradient Boosting |
| SVM | Support Vector Machine |
| AdaBoost | Adaptive Boosting |
| FFT | Fast Fourier Transform |
| RMSE | Root Mean Square Error |

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
