# Peer review of "AdaBoost Ensemble Approach with Weak Classifiers for Gear Fault Diagnosis and Prognosis in DC Motors"

_applsci, doi:10.3390/app14073105_

Round 1

Reviewer 1 Report

Comments and Suggestions for Authors

This paper proposed a novel predictive methodology for diagnosing and predicting gear problems in DC motors. Some popular algorithms are discussed and applied to the system for comparison. There are still some issues to be clarified before the paper is published.

1.     Is the data source described in this sentence? “In an 8000-millisecond timeframe, which accommodates approximately 8 starting attempts of 1 second each, the motor’s operational cycles are observed, reflecting the dynamic engagement of the starter gear with the engine flywheel.”

If so, the way it is obtained seems to be different from general data sets. Is this reasonable?

2.     What proportions of training, testing and validation are in the experiments?

Reviewer 2 Report

Comments and Suggestions for Authors

1.

There is nothing written about the fauls in DC motor gear systems.

So, nobody can know what faults are assumed.

There is also no mention of information of the DC motor and gear system at all.

After reading this paper, readers will not be able to reproduce the experimental results.

2. Units in several figures

There are no units in a vertical and horizontal axes in several figures.

Therefore, these figures are meaningless because the units are not shown. 

I hope the authors confirm them carefully. 

3. line 430

The frequency spectrum for load current reflects the state of the machine.

But, the relationship between features extraced from spectrum and faults is not clear. You should explain why certain frequencies appear.

Furthermore, if the authors use frequency components between 0.1-0.2Hz, these results should be shown in frequency spectrum in Figure.4.

4.

I don't know how to read Tables.

5.

According to conclusion, it is said that AdaBoost with Decision Trees is a promising technique for fault diagnosis and prognosis in DC motor gear systems.

However, this result seems to be applicable only to the experimental setup used this manuscript.

Reviewer 3 Report

Comments and Suggestions for Authors

This study introduces a prediction method for diagnosing and predicting faults in DC motors. Using AdaBoost's weak classifier and regressor, the diagnosis aspect classifies the current operating status of the machine by analyzing the time-frequency features extracted from the motor current signal. The article is generally interesting, but there are some problems in the literature review, comparison of results, and innovation. I only agree that this paper may be published after revision. Here are some specific suggestions:

1. It is recommended that authors summarize the contribution of the method proposed in this article in the introduction of the article to highlight the research advantages of this article.

2. Regarding the literature review part of the article, some analysis work on RUL prediction is relatively old. It is recommended that the authors analyze some recent related work, such as multi-hop graph pooling adversarial network for cross-domain remaining useful life prediction: A distributed federated learning perspective

3. The article introduces many methods in the second part, such as k-means clustering, linear regression, long short-term memory network, random forest classifier, gradient enhancement machine, extreme gradient enhancement, logistic regression classifier, support vector machine, linear Kernel, decision tree classifier, and AdaBoost weak classifier, etc. These are methods that already exist and are widely used. This article only integrates methods, which does not seem to be innovative.

4. Although the article has tested many types of methods, it lacks comparison of methods, especially comparison with other literature. You can refer to some recent work, such as lifetime extension approach based on levenberg-marquardt neural network and power routing of dc-dc converters

5. Although the article introduces many algorithms, it lacks theoretical contribution and innovation, and does not have its own core method.

6. Some pictures in the article are not clear enough. It is recommended to modify the format of the pictures to increase the readability of the paper.

7. It is recommended to add pseudo code of the algorithm and summarize it.

Comments on the Quality of English Language

Please see the comments to the authors.

Round 2

Reviewer 3 Report

Comments and Suggestions for Authors

Thanks the authors' revision. I accept its publication.

Comments on the Quality of English Language

Thanks the authors' revision. I accept its publication.